# Fractal Dimension and Texture Analysis of Lesion Autofluorescence in the Evaluation of Oral Lichen Planus Treatment Effectiveness

**DOI:** 10.3390/ma14185448

**Published:** 2021-09-21

**Authors:** Kamil Jurczyszyn, Witold Trzeciakowski, Marcin Kozakiewicz, Dorota Kida, Katarzyna Malec, Bożena Karolewicz, Tomasz Konopka, Jacek Zborowski

**Affiliations:** 1Department of Oral Surgery, Wroclaw Medical University, Krakowska 26, 50-425 Wroclaw, Poland; 2Institute of High Pressure Physics, Polish Academy of Sciences, 01-142 Warsaw, Poland; witold.trzeciakowski@unipress.waw.pl; 3Department of Maxillofacial Surgery, Medical University of Lodz, 113 S. Zeromski Street, 90-549 Lodz, Poland; marcin.kozakiewicz@umed.lodz.pl; 4Department of Drugs Form Technology, Wroclaw Medical University, Borowska 211A, 50-556 Wroclaw, Poland; dorota.kida@umed.wroc.pl (D.K.); k.malec@umed.wroc.pl (K.M.); bozena.karolewicz@umed.wroc.pl (B.K.); 5Department of Periodontology, Wroclaw Medical University, Krakowska 26, 50-425 Wroclaw, Poland; tomasz.konopka@umed.wroc.pl (T.K.); jacek.zborowski@umed.wroc.pl (J.Z.)

**Keywords:** photodynamic therapy, topical steroid, oral lichen planus, fractal dimension, texture analysis, autofluorescence

## Abstract

Background: Oral Lichen planus (OLP) is a chronic inflammatory disease. Topical steroids are used as the treatment of choice. The alternative is photodynamic therapy (PDT). The study aimed to fabricate optimal biodegradable matrices for methylene blue or triamcinolone acetonide because of a lack of currently commercially available carriers that could adhere to the mucous. Methods: The study was designed as a 12-week single-blind prospective randomized clinical trial with 30 patients, full contralateral split-mouth design. Matrices for steroid and photosensitizer and laser device were fabricated. Fractal and texture analysis of photographs, taken in 405, 450, 405 + 450 nm wavelength, of lesions was performed to increase the objectivity of the assessment of treatment. Results: We achieved two total responses for treatment in case of steroid therapy and one in the case of PDT. Partial response was noted in 17 lesions treated using local steroid therapy and 21 in the case of PDT. No statistically significant differences were found between the effectiveness of both used methods. Statistically significant differences in fractal dimension before and after treatment were observed only in the analysis of photographs taken in 405 + 450 nm wavelength. Conclusions: Photodynamic therapy and topical steroid therapy are effective methods for treating OLP. Using a carrier offers the possibility of a more predictable and effective method of drug delivery into the mucous membrane. Autofluorescence enables the detection of lesions especially at the early stage of their development.

## 1. Introduction

Lichen planus is a chronic inflammatory disease with an unclear aetiology that involves stratified squamous epithelium of the skin, mucous membranes and reproductive organs [1]. The prevalence of lichen planus in the general population is estimated at up to 5% and that of its oral form at 0.5–2.2% with a female to male incidence ratio of 2:1 [2]. The disease most frequently affects individuals over 30 years old. Symptoms within the oral cavity develop in as many as 60% of patients with their skin affected by the disease. On the other hand, only 15% of patients in whom the disease is limited to their mouth develop symptoms involving the skin or reproductive organs [3]. Clinically, lichen can have an appearance ranging from asymptomatic white keratotic lesions to forms involving erosions and ulcers, most frequently, in 14.9% of cases, located symmetrically in the area of the cheeks [4]. According to Andreasen, six forms of oral lichen planus (OLP) can be distinguished: the reticular, plaque, atrophic, papular, erosive and bullous form [5]. In 2005, the WHO (World Health Organization) classified oral lichen planus as a potentially premalignant lesion with a capacity to transform into squamous cell carcinoma [6]. In an analysis of 16 studies, the incidence of malignant transformations of OLP ranged from 0 to 3.5% with a mean of 1.09% [7]. Despite a long tradition of research on lichen, its aetiology is not fully explained. Idiopathic lesions and lesions related to additional factors modifying the immune system can be distinguished. Lichenoid lesions most frequently develop as contact allergies to amalgam dental fillings containing mercury as manifestations of a delayed-type hypersensitivity reaction. These lesions can also appear as a response to ACE (angiotensin-converting enzyme) inhibitors and beta-blockers (antihypertensive drugs), NSAIDs (nonsteroidal anti-inflammatory drugs), and antimalarial medications used in rheumatoid arthritis. Lichenoid tissue reactions can also develop in bone marrow transplant patients as a symptom of graft versus host disease (GVHD) [8].

Several hypotheses have been proposed to explain the aetiology of lichenous lesions. They are, first of all, related to immune regulatory disorders and increased CD4 (cluster of differentiation 4) and CD8^+^ (cluster of differentiation 8) T cell reactivity, mast cell degranulation. In patients with OLP, there is an increased release of proinflammatory mediators such as IFN-g (Interferon gamma) and TNF-α (tumour necrosis factor α). Expression levels of MMP (matrix metalloproteinases), especially of MMP-9, are also disturbed. They are much higher in OLP than in healthy controls. The production of proinflammatory chemokines, such as RANTES (Regulated on Activation, Normal T-cell Expressed and Secreted), also increases [9]. Genetic predisposition related to increased expression of HLA-DR1(Human Leukocyte Antigen—DR isotype), DQ3, DR3 or DR9 antigens is a significant cause of OLP, nonetheless, the immunogenetic background of OLP has not been determined so far [10]. It has been found that there is a strong correlation between HBV/HCV (hepatitis B virus, hepatitis C virus) and the prevalence of OLP. This relationship is explained by the capacity of the virus to replicate not only in hepatocytes but also in the cells of the skin and mucous membranes. It has been proven that the risk of OLP in patients with HCV is twice as high as in the general population. The prevalence of lichen is also connected with viral infections such as HPV (human papillomavirus) 16 and 18, EBV (Epstein–Barr Virus) and CMV (Cytomegalovirus) [11]. A high level of stress is often mentioned among the causative factors of OLP. Patients associate the occurrence of lesions or the exacerbation of symptoms of lichen with periods of increased nervous tension. This relationship results from elevated cortisol levels, an imbalance between Th1 and Th2 cytokines caused by increased secretion of neuroendocrine hormones during exposure to stress [12]. According to the current state of the art, there is no single most effective method for treating OLP. The alleviation of pain experienced by patients remains the primary objective of the treatment. The form of the therapy, the medication used, and the duration of its action are primarily related to the clinical type of the lichenoid eruption. Patients should avoid direct mechanical irritation of the lesions and modify their diet to eliminate individual inducing factors. OLP is currently treated with corticosteroids, retinoids, immunosuppressants (e.g., tacrolimus) or immunomodulators (e.g., cyclosporine) [13]. Topical steroids are used as the treatment of choice for OLP lesions as a widely recognized and used method which, however, causes a number of adverse drug reactions. They include, but are not limited to, fungal superinfections, mucosal atrophies, recurrence of lichenoid lesions and xerostomia. The greatest disadvantage of topical corticosteroids is their nonadherence to the mucous membrane which results in relapses of the disease and disappointment with treatment effects. Systemic glucocorticoids are reserved for patients in whom topical treatment has failed or those diagnosed with OLP with cutaneous, genital, oesophageal and/or scalp involvement [14].

The necessity to search for alternative treatment methods is self-evident. Photodynamic therapy, whose effects are similar to those obtained with the use of conventional treatment methods, seems a particularly promising solution for patients not responding to treatment or those in whom steroids are contraindicated. Its main disadvantages include problems with applying the photosensitizer and keeping it in direct contact with the treated lesion, which in our study was eliminated with the use of adhesive carriers with an active agent [15].

Photodynamic therapy (PDT) involves two main inseparable agents: light and photosensitizer (PS). It is important to underline that the doses of light and photosensitizer are too low to effectively act separately. PDT is used in the case of non-malignant and malignant lesions. Photodynamic therapy is widely applied in dermatology in actinic keratosis, acne vulgaris or basal cell carcinoma [16,17,18,19]. In the case of skin lesions, the application of PS is relatively easy. In most cases, photosensitizer is in the ointment form, and the accumulation time of PS may be long. In patients with oral lesions, the situation is more complex. The oral cavity environment is totally different from the skin environment with the continuous secretion of saliva as its main disadvantage. Application of PS is difficult, and time of PS accumulation is significantly lower than in skin lesions due to saliva secretion. In our study, we tested the application of a novel mucous adhesive carrier of PS and steroids. 

Another problem is the objective estimation of the effectiveness of therapy in the oral cavity. The size of lesions is the most intuitive factor which may be applied to describe the effects of treatment. In oral cavity, lesions are situated in soft tissues such as mucous membranes of cheeks or tongue. In such locations, their size strongly depends on muscle tension so such a way of estimation of treatment effectiveness may fail. Another disadvantage is the irregular shape of lesions. In our study, we used fractal dimension analysis (FDA) and texture analysis (TA) of lesion images to increase the objectivity of the assessment of treatment. In Euclidian geometry, we are used to the fact that dimension is an integer, for example the dimension of a point is 0, section has one dimension—length, whereas length and width describe flat figures and solids possess three dimensions: length, width and height. Fractals go beyond those simple rules. Their dimensions are rational numbers and may take values between 0 and 3. For example, if we analyse shape, the fractal dimension will be lower if complexity of shape will increase. In our study, we used the intensity difference modified box-counting method to calculate fractal dimension which enabled us to analyse grey scaled images. FDA is applied in some of studies on the analysis of irregular shapes such bone structure of computed bone tomography and radiographic images or cancer blood vessels [20,21,22,23].

Another useful mathematical method which is helpful in surface description is texture analysis (TA). All digital images consist of pixels. Each pixel is described by two features: coordinates and colour/brightness. Texture is a collection of recurrent graphical patterns characterized by brightness, entropy, smoothness, uniformity, roughness, granulation, randomness, or linearity [24]. TA offers the possibility of surface analysis. Texture analysis is widely applied in the case of magnetic resonance, computed tomography or X-ray images [25,26,27].

There are a few commercial methods for diagnosing lesions of the oral mucosa based on the autofluorescence of the lesions in various wavelengths. In some cases, it is blue or violet light [28].

Our study aimed to compare the effectiveness of photodynamic therapy with topical steroid therapy in OLP with the use of novel carriers for the photosensitizer and steroids. To assess the effectiveness of both methods, fractal dimensions and texture of lesion images obtained in various wavelengths were analysed. 

The null hypothesis was that there were no significant differences in the effectiveness of photodynamic therapy and steroids therapy in the aspect of lesions’ size, fractal dimension and texture features in various wavelengths.

## 2. Materials and Methods

### 2.1. Patients and Lesions

The study was designed as a 12-week single-blind prospective randomized clinical trial with the full contralateral split-mouth design conducted in patients with bilateral erythematous or erosive oral lichen planus. The split-mouth RCT (randomized controlled trials) method was chosen because the authors aimed to reduce the influence of numerous variables on the effectiveness of the topical treatment of OLP between groups of patients. The required sample size was calculated to compare the proportions of two paired samples (McNemar test formula was applied) [29] where: type I error rate was 5%, power was 0.8, the proportion of success in both groups 0.75, and the proportion of failure in both groups 0.25. The size of the study sample was 30. Thirty patients participated in the trial. Four patients discontinued PDT, one patient discontinued topical corticosteroid therapy without discontinuing the treatment to the contralateral side, and two patients did not attend the final examination. Ultimately, the results of OLP treatment were analysed in 28 patients and, with respect to eruptions, 24 were treated with PDT and 27 with TA.

Local adverse reactions to the applied therapy were observed only in the 9 days of active treatment. In 4 patients, after the first or second PDT procedure, inflammatory OLP lesions were aggravated, mild oedema and stronger pain were observed, resulting in the patients’ refusal to continue the therapy on the affected side of the mouth. One of the elderly patients resigned from unassisted administration of the polymer carrier with TA as a result of technical problems with the application of the drug. One patient reported increased halitosis in connection with the treatment. No general adverse reactions were observed during the treatment period or during postoperative observation. It was not determined why two of the treated patients did not attend the final follow-up.

The main inclusion criterion was a clinical diagnosis of bilateral erythematous or erosive OLP, which was confirmed with histopathological results of punch biopsy collected from the most clinically relevant site. We applied the following exclusion criteria: dysplasia on histopathological examination, liver disease, diabetes, graft-versus-host disease, nicotinism, pregnancy, breastfeeding, hypersensitivity type IV. The patient had to have bilateral OLP lesions confirmed by histopathological examination greater than 10 mm in size requiring topical treatment. 

The surface area of the clinical lesion was assessed with the use of a PCP UNC15 periodontal probe (Hu-Friedy, Chicago, IL, USA) with markings every 1 mm (the maximum length and width of the lesion were measured and used for calculating its surface area in mm^2^) and the scale for the evolution of OLP lesions developed by Thongprasom et al. [30]. The Thongprasom scale is on a 5-point scale, where: 0—no lesion, normal mucous membranes, 1—mild white striae, no erythematous areas, 2—white striae with an atrophic area <1 cm, 3—white striae with an atrophic area >1 cm, 4—white striae with an erosive area <1 cm, 5—white striae with an erosive area >1 cm.

Lichenoid lesion was diagnosed histologically following histological features that included irregular acanthosis, degeneration of the basal layer of the epithelium and a band of lymphohistiocytic infiltrate in the upper chorion composed almost exclusively of mature lymphocytes [31]. 

All procedures were conducted after obtaining the approval of the Ethics Committee of Wroclaw Medical University, Poland (approval No. KB 845/2020—8 February 2021) and was registered in clinicaltrials.gov, accessed on 17 September 2021, with the number NCT04991012.

### 2.2. Laser Device

In our study, we used a 405 nm, 450 nm and 640 nm laser device. This device was invented and produced at the Institute of High Pressure Physics, Warsaw, Poland. A reflector in the form of a regular pyramid couples laser diodes to a multi-mode fiber. This device is able to couple into one optic fiber two or three wavelengths together. In this study, we used two wavelengths: 405, 450 and 405 + 450 nm together for the autofluorescence of lesions, and 640 nm for photodynamic therapy.

### 2.3. Porous Matrices Preparation and Evaluation

#### 2.3.1. Methylene Blue Porous Matrix Preparation

Freeze-drying technique was used to prepare porous matrices. Polymers were added to purified water to obtained separately solutions of pullulan 15% *w*/*w* (abcr, Karlsruhe, Germany) and alginate acid sodium salt 4% *w*/*w* (Sigma, Steinheim, Germany) respectively. Compounds were homogenized through mechanical stirring (L366, Labinco BV, Breda, The Netherlands) with stirrer speed fixed to avoid air bubbles. Forward, glycerol 95% (Sigma, Steinheim, Germany), methylcellulose (4000 cp, Sigma, Steinheim, Germany), and methylene blue (Alfa Aesar, Kandel, Germany) were added to gain the concentrations in dry mass according to Table 1. Then mixture was aeration and homogenized by mix and whipping at 2340 rpm (Gako e/s, Eprus, Bielsko-Biała, Poland) for 10 min to get a foam, 40.0 g of mixture was transferred to a Petri dish (teflon-coated) and was frozen at −26 °C for 48 h and then freeze-dried at room temperature with ultimate vacuums of 9.0 × 10^−2^ to 1.3 × 10^−1^ mBar (Lyovac GT2, Steris, Köln, Germany) for 20 h. Matrices after drying are depicted on Figure 1a,b. were preserved from moisture and light at ambient temperature for another characterization.

#### 2.3.2. Steroid Porous Matrices Preparation

The porous matrices for triamcinolone acetonide application (Figure 2) were fabricated initially without active substance analogously to the described above method for the preparation of the matrix with methylene blue. The physicochemical properties of matrices without steroids according to the composition presented in Table 2 were evaluated after being carefully peeled off from plates. For triamcinolone acetonide application (Carbosynth Limited, Compton, Berkshire, UK), optimal porous formulation Pul40/Alg was chosen. The matrix with 0.05% triamcinolone acetonide in dry mass was prepared analogously to optimal placebo composition, adding active substance to mixing formulation before poured and lyophilisation.

#### 2.3.3. Mechanical Properties of Testing Matrices 

TA.XTplus Texture Analyser (Stable Micro System, Godalming, UK) was used to evaluate mechanical properties of matrices in an ambient temperature. Apparatus was associated with two clamps; upper and lower clamp (tensile grips Type A/TG), former was fixed and later was free to move. Rectangular segments (4 cm length, 1 cm width) of matrices sample were chosen and held by both clamps at an initial separation of 10 mm. After force was applied till the breakage of matrix, the maximum force before break estimated and percentage elongation of the matrices calculated as follows: $\%\;Elongation=\frac{LB}{IL}\times 100$, where *LB* is final length and *IL* is initial length of sample film.

#### 2.3.4. Porous Matrices Disintegrating Test 

The matrices disintegration was represented as the durability of the matrix structure during dissolution process in medium in time. Samples with a size of 2.5 cm × 2.5 cm with closely similar masses were immersed in 10 mL of water in the case of matrices with methylene blue and in 10 mL of PBS for matrices with triamcinolone acetonide, and incubated at 37 °C till 300 min in a sealed closed vessel with horizontal shaking 20 times per minute. The test was proceeded for 400 min for Pul20/Alg formulation. The test was repeated six-times for each formulation. Stages of matrices disintegration are shown on Figure 3.

#### 2.3.5. pH Studies 

The pH value of the extracts after blurring the matrices was determined potentiometric method. The film samples 2.5 cm × 2.5 cm were transferred to the flask with distilled water (10 mL) and mixed to obtain an aqueous extract. Next, a pH meter electrode (ERH-11, Hydromet, Gliwice, Poland) was immersed into the extract and allowed to equilibrate for 1 min, and the pH was noted. 

#### 2.3.6. UV-VIS Method Methylene Blue Determination 

Basal solution in the concentration of 0.2% methylene blue (MB) was prepared in 100 mL of distilled water. Standard solutions were prepared in orange flasks in concentrations respectively of 0.003 mg/mL, 0.005 mg/mL, 0.01 mg/mL, 0.015 mg/mL, 0.02 mg/mL and 0.03 mg/mL methylene blue. The absorbances of the solutions were measured using the UV-Vis JASCO V-650 double-beam spectrophotometer (JASCO, Tokyo, Japan) at λ = 589 nm. The calibration curve of methylene blue showed linearity in the predicted concentration range with regression value of 0.994. The equation (y = 20.914 x) was derived from of the methylene blue absorbance against respective drug concentration. This UV-VIS method was used to determine the drug contents in samples of prepared matrices as well as drug release from these matrices.

#### 2.3.7. In Vitro Methylene Blue Release Studies 

The fragments of polymer matrices (2.5 cm × 2.5 cm) were inserted into three 50 mL orange volumetric vessels, then 10 mL of distilled water was added. The samples were placed in shaking water bath at a temperature of 37 °C, set shaking 60 times per minute (at with horizontal). The samples were protected with parafilm and aluminium foil against water evaporation and sunlight. The samples to methylene blue determinations were taken at 16 time points to 195 min. From each of the three vessels, 200 µL samples was taken and water diluted to 625-fold finally. After samples taken, to all vessels was added equivalent water volume with the temperature of 37 °C. Based on calibration curve, the amount of release methylene blue by time was calculated.

#### 2.3.8. HPLC Method of Triamcinolone Acetonide Determination

A series of triamcinolone acetonide concentrations to obtain a calibration curve were prepared by dissolving the drug in methanol:water mixture 75:25 *v*/*v* ranging from 1 to 100 μg/mL. These solutions were analysed using Agilent 1260 Infinity Quaternary System HPLC (Agilent Technologies Ltd., Stockport, UK) connected to UV/Visible spectrophotometer which was set at 240 nm. A reverse phase Thermo Scientific Hypersil Gold C18 column (150 mm × 4.6 mm, 5 μm, Thermo-Scientific, Waltham, MA, USA) held at 30 °C was used for the chromatographic separation. The mobile phase consisted of a mixture of 50:50 *v*/*v* water:acetonitrile and was delivered with steady flow rate of 0.8 mL/min. The injection volume was 10 μL. The calibration curve of triamcinolone acetonide showed linearity in the predicted concentration range with regression value of 0.999. The equation (y = 24.688x − 7.4975) was derived from the areas of the triamcinolone acetonide absorbance peak plotted against respective drug concentration. This HPLC method was used to determine the drug contents in samples of prepared matrices as well as drug release from these matrices.

#### 2.3.9. Triamcinolone Acetonide Content Uniformity in Porous Matrix

Drug content uniformity within optimized batch was determined by extracting the drug from matrices. Individually weighed sample (25 mm × 25 mm) of matrix was immersed (n = 3) in methanol in an orange volumetric flask and stirred for 24 h using Magnetic stirrer (IKA^®^ Poland Sp. z o.o., Warszawa, Poland). An aliquot of the filtrate was analysed for triamcinolone acetonide content using reverse phase HPLC with reference to a previously constructed calibration curve, and results carried to relative to the declared content in formulation.

#### 2.3.10. In Vitro Triamcinolone Acetonide Release Studies 

Samples of the triamcinolone acetonide loaded matrix, with a size of 25 mm × 25 mm with closely similar mass, were immersed in 15 mL water and incubated at 37 °C in a sealed closed vessel and wrapped in aluminium foil with horizontal shaking 60 times per minute. Accurately 1.5 mL of each sample was withdrawn at predetermined time intervals (30, 60, 90, 120, 150, 180, 210, 240, 270, 300, 330, and 360 min) filtered through 0.45 micron membrane filter and analysed to determine the of drug concentration in samples using HPLC. 

### 2.4. PDT Procedure

The treated lesion was entirely covered for 10 min with a carrier containing of methylene blue and then irradiated with a diode laser with a wavelength of 640 nm, at a dose of 120 J/cm^2^ and a power of 520 mW, spot diameter 8mm, power density 1.034 W/cm^2^, irradiation time 227 s. This procedure was always conducted by the same physician (KJ). Photodynamic therapy was repeated in three sessions, at three-day intervals.

### 2.5. Steroid Application

The qualified contralateral OLP lesion was treated by attaching to it, for 8 consecutive days, a carrier with 0.05% triamcinolone acetonide cut to the size of the lesion. On the days on which PDT was administered, at the end of such a session, the patient had a steroid drug attached to the lesion on days 2, 4, 5 and 7 starting from treatment commencement. The carrier which was cut to the size of the lesion was self-administered by the patients after they brushed their teeth in the evening. Both forms of treatment were applied simultaneously. 

### 2.6. Image Acquisition

All photographs were taken using a Canon EOS 77D, Canon 60 mm f/2.8 EF-S USM Macro lens (Canon, Ōta, Tokyo, Japan) with Metz 15 MS-1 ring light (Metz, Markham, ON, Canada). 

The same distance (45 cm) was achieved to take all of photos. An optical axis of the camera was kept perpendicular to the surface of the lesion. The focus plane was locked to 45 cm to obtain repeatability. Photographs during laser irradiation were taken in the same way as in the white light. Laser parameters were set to 0.8 W for single wavelengths (405, 450 nm), and 1.6 W for 405 + 450 nm. The optic fibre was equipped with a special diffuser of light which enables uniform irradiation of the surface. To achieve similar histogram filling, we used the following parameters of photographs: 405 nm—ISO 3200, f/9, time of exposure 1/90 s, 450 nm—ISO 3200, f/9, 1/250 s, 405 + 450 nm—ISO3200, f/9, 1/500 s. Examples of photographs taken in various wavelengths are shown in Figure 4.

### 2.7. Fractal Dimension Analysis

An ImageJ version 1.53e (Image Processing and Analysis in Java—Wayne Rasband and contributors, National Institutes of Health, Bethesda, MD, USA, public domain license, https://imagej.nih.gov/ij/, accessed on 26 July 2021) and the FracLac plugin version 2.5 (Charles Sturt University, Bathurs, Australia, public domain license) was used to perform all fractal analysis. 

We applied the modified algorithm to the counting box. This method allows us to analyse monochromatic images. We applied the intensity difference fractal dimension counting method. In this algorithm, the analysed image is divided into boxes like in the classical counting box method. The difference between maximum pixel intensity and minimum pixel intensity is counted in each box (δI_i,j,ε_, where: (δI—the difference between maximum pixel intensity and minimum pixel intensity i, j—location of the analysed box in a scale ε):δI_i,j,ε_ = maximum pixel intensity_i,j,ε_ − minimum pixel intensity_i,j,ε_(1)

In the next step, 1 is added to the intensity difference to prevent its value to be 0:I_i,j,ε_ = δI_i,j,ε_ + 1(2)

Finally, fractal dimension of the intensity difference is described by the following formula:(3)$$\mathrm{FD}=\underset{\varepsilon \to 0}{\lim}\frac{\ln\left(I\varepsilon \right)}{\ln\left(\frac{1}{\varepsilon }\right)}$$
where: FD—fractal dimension of intensity difference, I_ε_ = Σ[1δI_i,j,ε_ + 1], ε—scale of box.

All operations are shown in Figure 5.

The region of interest (ROI) for fractal analysis was set at 200 × 200 (gingival lesions) and 300 × 300 pixels (cheeks and tongue) it depends on the lesion size. All colour images of ROIs were converted and saved as 8-bit monochromatic bitmaps. GIMP version 2.10.24 (GNU Image Manipulation Program—www.gimp.org, accessed on 26 July 2021, free and open-source license) was used to apply graphical operations. 

### 2.8. Texture Analysis

The texture of oral mucosa was evaluated using features derived from co-occurrence matrix [33,34,35]. All ROIs were normalized (μ ± 3σ) to obtain the same average (μ) and standard deviation (σ) of optical density. Entropy and difference entropy from the co-occurrence matrix a well short- and long-run emphasis moment from the run-length matrix in ROIs were calculated for each composite material tested:(4)$$\mathrm{Contrast}=\overset{\mathrm{Ng}-1}{\underset{\left|i-j\right|=0}{\sum }}{\left(\left|i-j\right|\right)}^{2}\overset{Ng}{\underset{i=1}{\sum }}\overset{\mathrm{Ng}}{\underset{j=1}{\sum }}p\left(i,j\right)$$
where Σ is the sum, Ng is the number of optical density levels in the radiograph, i and j are the optical density of pixels that are 5 pixels away from one another, p is probability, and log is the common logarithm. Rather, it is a measure of microcontrast because the image is sampled at distances of 5 pixels. 

### 2.9. Statistical Analysis

Statistica version 13.3 (StatSoft, Cracow, Poland) and Stargraphics Centurion 18 ver.18.1.12 (StarPoint Technologies, Inc., Addison, VA, USA) were applied to calculate all statistical tests and 0.05 was set as the statistical significant level. The normality of distribution was confirmed by The Shapiro–Wilk test. Due to normal distribution, we performed parametric tests. We applied the Student *t*-test for paired (check for differences between lesion size before and after treatment) and unpaired samples (estimation of differences in FD between lesion before and after treatment and lesion versus healthy mucous membrane). The correlation matrix was applied to calculate the correlation coefficient between FD of images in various wavelengths and the surface size of lesions before and after treatment, the difference between the size of lesions, percentage of increase in the lesion and Thongprasom scale.

## 3. Results

We achieved two total responses for treatment in case of steroid therapy and one total response in case of PDT. Partial response was noted in 17 lesions treated using local steroid therapy and 21 in case of PDT. Enlarge in lesion size was observed in five lesions treated using steroid and in two lesions in the case of PDT.

Table 3 presents a comparison of medium-sized lesions before (Surf0) and after the treatment (Surf1) depending on the treatment method. A paired Student t-test was performed. Statistically significant differences were found between the surface area of lesions before the treatment in relation to their surface area after the treatment for both treatment methods. The mean surface area of the lesions treated with a steroid was initially 11.4 mm^2^ and was reduced after the treatment to 7.1 mm^2^, with a difference of 4.3 mm^2^. In the case of lesions treated with PDT, the initial surface area of the lesions was 18.4 mm^2^ and was reduced after the treatment to 11 mm^2^. In lesions treated with steroids, the surface area was reduced by 4.3 mm^2^ as compared to those treated with PDT whose surface area was reduced by 7.3 mm^2^. Table 4 presents the results of the unpaired Student’s t-test which did not reveal any statistically significant differences between the value of the surface area of the lesions after the treatment with the use of both methods, which leads to the conclusion that both methods are equally effective.

Table 5 presents the values of fractal dimension (FD) of the lesions before and after the treatment and the reference mucous membrane in white, blue (450 nm), violet (405), and blue and violet (405 + 450 nm) light together. The lowest value of FD of the lesions before the treatment was recorded for the combined use of two wavelengths 405 + 450 nm (1.473). On the other hand, the highest value of FD of the lesions was recorded for white light (1.589). A statistically significant difference was found between the values of fractal dimension of the lesions as compared to the reference mucous membrane.

Of note, despite the statistically significant differences in the changes to the surface area of the lesions before and after the treatment, we have not found such changes in terms of FD, except for the lesions analysed with combined wavelengths of 405 nm and 450 nm.

Table 6 presents a summary of the correlation coefficient (r). A moderate negative linear relationship (r = −0.45) was found between the surface area of the lesion before the treatment and its fractal dimension calculated for blue light. For lesions treated with photodynamic therapy, a moderate negative linear relationship was found between fractal dimensions calculated for blue light (r = −0.598) and a relatively strong relationship for violet illumination (r = −0.748).

A relatively strong negative linear relationship was also observed between FD of lesions illuminated with violet light and the percentage of lesion reduction (r = −0.837).

For lesions treated with steroids, a moderate negative linear relationship was recorded (r = −0.580) between FD of the lesion in white illumination compared to the difference in its surface area after and before the treatment. A weak linear relationship was also observed between FD in violet illumination and the percentage of lesion reduction (r = 0.358).

Texture analysis results are presented in Figure 6 together with original intra-oral photography with separation to four illumination techniques.

Each of the ways of illuminating the lesion can be easily distinguished from the normal oral mucosa as microcontrast of image texture is analysed (*p* < 0.001). It should be noted, however, that lichen planus lesions are most reliably distinguished in blue+violet light (*p* < 0.000005), followed by blue light (*p* < 0.0001) (Table 7). Lichen lesions are characterized by significantly lower microcontrast in all four types of illumination. The changes that occurred in lesions under the two treatments were checked intraorally under four illumination conditions as well. 

There were no statistically significant differences in lesion texture between the two treatment groups (Table 8). The lesion texture was the same in both groups at baseline (pre-treatment). An improvement in the texture of the treated lesion was noted only after steroid therapy by examining the lesion appearance under blue+violet light. In general, it can be seen that none of the treatment methods used unfortunately transforms the lesion to a textural appearance similar to that of normal mucosa. The relative superiority of steroid therapy over PDT can be seen in the blue-violet observations, where the microcontrast of the treated lesion significantly increased after the treatment cycle. Under white light, the lesion achieved the texture of normal mucosa (Control) after both steroid therapy as well PDT.

The appearance of lesions of the mucosa covering the tongue in blue and violet light showed the most pathological features in texture analysis (*p* < 0.05) while lesions in the cheek mucosa had a texture closer to normal. Gingival pathologies had microcontrast intermediate between buccal and tongue. The dichotomous division into mucosa covered with keratinized and non-keratinized epithelium confirms the previous observation. Lichen localized in keratinized epithelium (tongue and gingiva) has more pathological features of microcontrast (i.e., lower value; *p* < 0.05) than lesions in non-keratinized epithelium (buccal mucosa; Figure 7). It should also be noted that this relationship is revealed not only in the blue+violet light, but also in the blue light itself (*p* < 0.05). Pathological severity (haematoxylin and eosin staining) are not associated with textural features observed in any type of light (as does the age of the patient, as well as the area affected by the primary lesion). Thongprasom’s classification grades are not related to the pathological microcontrast of the lesion.

The current study was aimed to fabricate optimal biodegradable matrices for methylene blue or triamcinolone acetonide to local dental application. The formulation compositions in Table 1 and Table 2 were successfully prepared by freeze dried technique. After physicochemical properties analysis, fabricated porous matrix with methylene blue was not very flexible, but was mechanically resistance with smooth, homogeneity surface. In time extends than 300 min of disintegration test formulation was partly resistance to disintegration process (see Figure 1c). The methylene blue cumulative release in 3 h time from matrix was over 80% of the declared value in carrier (see Figure 8A). After physicochemical properties fabricated porous matrices analysis as optimal formulation to triamcinolone acetonide application, the Pul40/Alg matrix was chosen. Porous formulation Pul40/Alg was elastic, tear resistance more than formulation Pul50/Alg and with smooth and soft surface. For a time greater than 360 min of disintegration test, the Pul40/Alg formulation was resistant to the dissolution process (see Figure 3). The triamcinolone acetonide cumulative release in time from Pul40/Alg matrix containing 0.05% substance in dry mass showed in Figure 8B. After 6 h test from matrix 58.74% of declared content of substance has been released. Table 9 shows physicochemical characteristics of the evaluated matrices.

## 4. Discussion 

Topical steroids are still the first choice in the treatment of oral lichen planus. They can cause adverse reactions such as fungal superinfections, mucosal atrophies, recurrence of lichenoid lesions and xerostomia [36]. Moreover, there are currently no commercially available carriers that could adhere to the mucous membrane to locally release the active agent and, consequently, provide more predictable treatment effects. In photodynamic therapy, an adequate concentration of the photosensitizer (PS) must be obtained in the treated lesion. In oral lesions, there is an environment rich in saliva which effectively leads to washing the PS off the treated lesion and, therefore, significantly reduces its concentration in the tissues. The innovative PS carrier offers opportunities to increase the effectiveness of PDT in oral lesions. The carrier, due to its high level of adherence to the mucous membrane, allows obtaining the optimal concentration of PS in the treated lesion. At the same time, if the steroid is washed with the saliva, which has adverse effects for the entire body and can be contraindicated in patients with other diseases, such as Diabetes Mellitus or hypertension [37]. It is difficult to compare treatment effectiveness due to the immense diversity of procedures. Red light can penetrate human tissues and with increasing wavelengths, it can reach its deeper layers. Most photosensitizers enable light to penetrate tissues from 0.5 cm (for 630 nm) to 1.5 cm deep (for ca. 700 nm). On this assumption, the appropriate photosensitizer is selected depending on clinical status and the treated pathology [38]. The most frequently used photosensitizers used in photodynamic therapy are 5% methylene blue, tolonium chloride and 5-ALA (5-aminolevulinic acid) which are applied mainly in the form of fluid, mouth wash or gel, which significantly influences the absorbability of the photosensitizer and results in the inability to control its concentration at the site of its administration [39]. In photodynamic therapy, light sources with different power, variable wavelength, energy density, time of irradiation, as well as the design of the study itself, make it difficult to directly compare treatment effects [40,41].

The conventional clinical assessment using the measurements of the surface area of lesions in white (full-spectrum) illumination revealed a statistically significant reduction in the size of lesions treated with photodynamic therapy and topical steroids. On the other hand, no statistically significant differences were observed in the effectiveness of both methods. Interestingly enough, there were also no statistically significant differences in the values of fractal dimension of lesions before and after the treatment, except for lesions assessed in light consisting of two components—405 and 450 nm. What is also interesting is the fact that a weak negative relationship was found between the size of the lesion before the treatment and its fractal dimension in blue illumination, and a weak positive relationship in full-spectrum light.

In their study of 20 patients with an erosive form of lichen planus, Salech et al. used methylene blue for a 5-min mouthwash to compare PDT with topical betamethasone. After 4 weeks of observation, despite the used application form, higher effectiveness was found for PDT than with treatment with topical corticosteroids [42]. Another split-mouth study compared the effectiveness of tolonium chloride to that of 0.1% triamcinolone acetonide in 11 patients. The photosensitizer was administered with a sterile gauze pad for 10 min and left on the mucous membrane in this form, after which photodynamic therapy was applied. The effectiveness of PDT was found to be similar to that of the steroid drug [43], just as in our study. In a systematic review, including a study completed in 2017, a problem was noted with regard to designing clinical trials with the use of photodynamic therapy. The review compared five clinical trials in which, despite significant differences in the discussed parameters—wavelength (320–660 nm), power density (130 mW/cm^2^) and exposure times (70–150 s) and a follow-up period between 4 and 48 weeks, it was found that the treatment effectiveness of photodynamic therapy is similar to that of topical steroids, but without a distinct advantage [40]. A review published in 2020 analysed studies published in the PubMed, Embase (Ovid) and Medline (Ovid) databases from 2006 to 2020 on the use of photodynamic therapy in the treatment of oral lichen planus. It was highlighted that the disadvantage of those studies was that only 5 out of 17 studies were designed as randomized controlled trials. Five different photosensitizers were analysed. The compared studies used lasers and electroluminescence diodes (LED) with wavelengths ranging from 420 nm to 682 nm. The review also revealed the effectiveness of photodynamic stimulation similar to that of topical administration of corticosteroids but, again, flaws in the design of the clinical trial were noted and it was concluded that photodynamic therapy seems a promising alternative to conventional treatment methods [44]. He et al., in their meta-analysis of five randomized studies which assessed the effectiveness of photodynamic therapy against 139 mucosal lesions, revealed that after the use of PDT the size of the lesions was reduced by 1.53 cm^2^ (95% confidence interval (CI): 0.71–2.35). The analysis of the available data revealed that 5-ALA was more effective than methylene blue (PR 0.87 (95% CI: 0.80–0.91), and it was noted that such a therapy seems a perfect alternative for patients in whom steroids cannot be used [41]. At the same time, Jajarm et al., in their study of 2014 which compared the effectiveness of photodynamic therapy using tolonium chloride as a photosensitizer with that of dexamethasone, obtained better clinical results in the treatment of OLP after the use of a steroid drug. It should be pointed out that in this study the photosensitizer was applied with a micropipette while the steroid was administered as mouthwash [45]. Photodynamic therapy, just as any treatment method, can result in post-treatment complications. The most frequently reported symptoms include burning, edema and cicatrization. However, compared to corticosteroids, they are definitely less significant [46]. Lavee, in his randomized clinical trial conducted in 2019, did not find any adverse reactions to PDT using tolonium chloride as compared to a group treated with a steroid drug [43]. Sulewska et al., in their study of 12 women with erosive lichen and a 12-month follow-up period, also emphasized the absence of significant adverse reactions after therapy with the use of 5-ALA as a photosensitizer [47].

The results of studies available in databases reveal that photodynamic therapy with all its advantages is becoming an alternative to the conventional form of treatment of lichen planus. Most importantly, it is minimally invasive and highly effective. Clinical improvement can be observed later than it is reported by patients, and the healing process continues for a long time after the completion of active treatment. Nevertheless, it is necessary to change the form of application of the active agent for obtaining its full effectiveness and achieving the clinical repeatability of the therapy. It would also be expected that a repeatable procedure protocol should be designed that would state the doses appropriate for their irradiation times enabling the direct comparison of treatment effects.

In all of the analysed wavelengths and in white light, the value of the fractal dimension of a healthy mucous membrane was always greater than the FD of the investigated lesions. Statistically significant differences were also found between the fractal dimension of the investigated lesions and healthy mucous membrane in all combinations of light. An analysis of fractal dimensions can be therefore helpful in the diagnosis of lichen planus. A moderate negative relationship between FD of a lesion in blue illumination and its size was observed. This result suggests that the larger the lesion, the smaller its fractal dimension. As was already mentioned, the smaller the fractal dimension of a lesion, the less it resembles healthy mucosa. The conclusion that can be drawn from the above is that the use of blue light offers an advantage over white light in the diagnosis of lichen.

In lesions treated with photodynamic therapy, a moderate negative relationship was found between FD of the lesions as illuminated with white and as illuminated with blue light, and a relatively strong negative relationship between lesions as illuminated with violet light and their surface area. On the other hand, such relationships were not observed with the use of topical steroids. In the case of steroids, only a weak positive relationship can be observed. On the other hand, in the case of lesions treated with steroids, a moderate negative relationship between the difference in surface areas of lesions before vs. after treatment and the value of their fractal dimension for white light. No such relationship could be observed for lesions treated with PDT. In the case of lesions treated with steroids, no relationship between FD and the Thongprasom scale was observed, while a moderate negative relationship with this scale was observed in the case of lesions treated with PDT and analysed in white and violet illumination. It therefore follows that the higher the value of fractal dimension in white and violet light, the lower the score on the Thongprasom scale. The clinical picture of lichen lesions of the oral mucosa comprises the white and red components. Wavelengths corresponding to blue and violet light are to a significant extent absorbed by hemoglobin (hyperemic/erosive areas of lesions) and melanin, observed as dark fields in photographs, and light within this wavelength band is well reflected by white areas of excessively keratinized epithelium. This relationship results in much more contrast in the picture of the lesions in the 405 + 450 nm band compared to full-spectrum light. In the market, there are diagnostic systems for oral mucosa using various wavelengths, however, none of them use two different wavelengths simultaneously. Our study reveals that statistically significant differences in the fractal dimension of lesions before and after treatment were observed only in the case of photographs in combined 405 + 450 nm light. Synergistic effect of two or three wavelengths is very interesting. In our previously study we revealed synergistic effect of two wavelengths 450 and 520 nm on increase in tissue temperature [48]. 

Assessment in white visible-spectrum light is characterized by low sensitivity. This causes difficulties in the clinical diagnosis of pathological lesions and correct assessment of their size, extent and severity—especially when their potential of malignant transformation is taken into account. Most importantly, it is difficult to detect them early and to adequately assess the effects and progress of their treatment. Autofluorescence is one of the techniques that could be used as a potential tool for assessing biochemical changes related to oral cavity disorders [49]. It enables the detection of lesions, especially at the early stage of their development. This method uses the absorption of particular wavelengths of light by fluorophores in the mucous membrane that can re-emit light as a result of fluorescence upon light excitation [50]. As early as 2006, Lane et al. used in their study a device of their own design to visualize fluorescence in tissues. The study was conducted on a group of 44 patients, and the device used blue light. The sensitivity of their method was assessed at 98%, and its specificity at 100% for differentiating healthy tissue from tissue with features of dysplasia [51]. Velcsope^™^ (Visually Enhance Lesion Scope) is a commercially available system that uses the phenomenon of autofluorescence. In a study conducted by Awam, 126 red and white oral cavity lesions were assessed. Reduced fluorescence was observed in 83% of lesions while the sensitivity of detection for dysplastic lesions was up to 84.1% [52]. Unfortunately, the use of solely Velscope or other techniques with the same operation principle can fail to detect areas of dysplasia but is of great significance in making decisions related to the type of the lesions, as well as in identifying biopsy sites. However, the weak specificity of the tool is the main limitation of its use as a tool for screening [53].

Contrast as a texture feature of oral mucosal images has been used previously [54]. Pathological solid keratinous masses blur the microcontrast structure of the mucosal surface.

It seems that observed in white light textural improvement is rather camouflage of limited effectiveness of both treatment methods, due to pessimistic results of microcontrast observed in blue and violet light. Any way some healing was reached as blue+violet light texture analysis shown i.e., removal of pathological of high contrast but without improvement the background of low contrast. Treated site is not similar to normal oral mucosa.

There are still no established treatment protocols for photodynamic therapy [55], therefore in our study we propose a scheme that could have clinical application in the treatment of OLP red lesions. We proposed a treatment protocol both in terms of the selection of power, light density, through the form of an adhesive carrier and the frequency of use. In addition, the proprietary form of administration used allows for more accurate management of the dose of the absorbed active agent, minimizing the time of application and increasing patient comfort. In addition, the analysis and evaluation of changes based on the OLP photographs taken in our study, compared to the standard, simple clinical evaluation method, may have a significant impact on the quality of treatment and the speed of recognizing potentially dangerous changes.

## 5. Study Limitations

As we mentioned in the introduction, delivery of drug in the environment of oral cavity is difficult. Adhesive carrier which we applied gives a possibility of more predicable way of drug delivery into mucous membrane. Another limitation of this study is taking a repeatable intraoral photography of lesions. In case of ling or cheek lesions variable tension of muscles may affect with shape of the lesion. We reduced this limitation using fractal and texture analysis. 

## 6. Conclusions

Photodynamic therapy and topical steroid therapy are effective methods for treating OLP.No statistically significant differences were found between the effectiveness of both used methods.Despite the significant reduction in lesions over the course of the treatment, statistically significant differences in fractal dimension before and after treatment were observed only in the analysis of photographs taken in 405 + 450 nm wavelength.In the 405 + 450 nm wavelengths pathological lesions in keratinized mucosa have significantly worse textural appearance than lesions formed in mucosa covered by non-keratinized epithelium.In spite of lack of statistical differences between effectiveness of PDT and steroids, we observed negative correlation coefficient between FD of lesion after treatment and the size of lesion after treatment. That correlation is not revealed for steroids treatment.Carriers of photosensitizer and steroid offers possibility of more predicable way of drug delivery into mucous membrane.

## 7. Patents

9:223:123, B2; date of patent: 29 December 2015.

## Figures and Tables

**Figure 1 materials-14-05448-f001:**
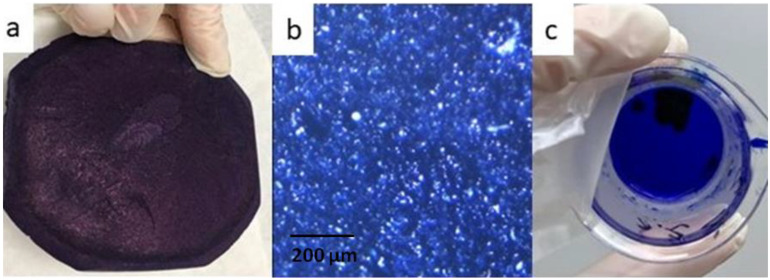
Images of methylene blue-loaded porous matrices (**a**) lower surface (picture taken with Nikon S9300 camera using macro lens); (**b**) microscope view at magnification 10 × cross section (picture taken with MB 200 FL—Opta-Tech (Warsaw, Poland), (**c**) fragment of methylene blue-loaded porous matrix after 90 min of the disintegration test with Canon EOS 77D, Canon 60 mm f/2.8 EF-S USM Macro lens (Canon, Ōta, Tokyo, Japan) with Metz 15 MS-1 ring light (Metz, Markham, ON, Canada).

**Figure 2 materials-14-05448-f002:**
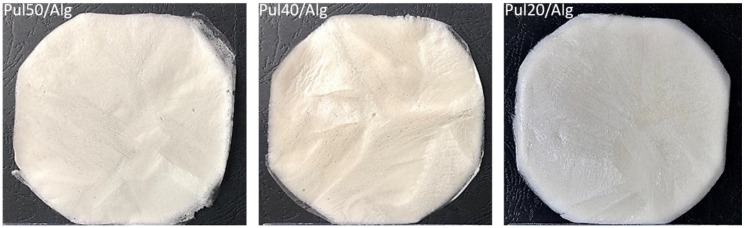
Optical images of porous matrices (picture taken with Nikon S9300 camera using macro lens).

**Figure 3 materials-14-05448-f003:**
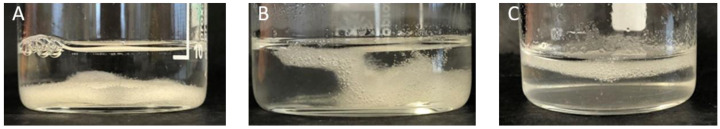
Stages of matrices disintegration (disintegration test of Pul40/Alg formulation in time: (**A**) matrices after 30 min, (**B**) matrices after 180 min, (**C**) matrices after 300 min of the dissolution process (picture taken with Nikon S9300 camera using macro lens).

**Figure 4 materials-14-05448-f004:**
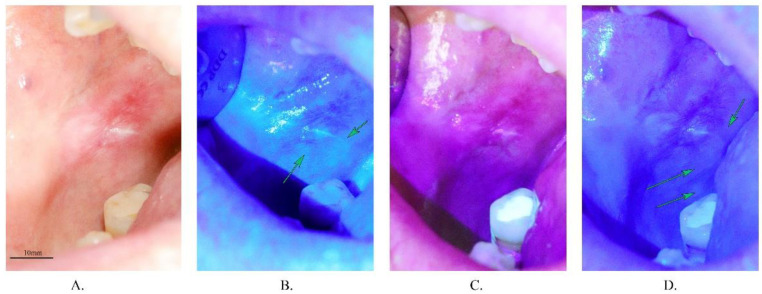
Examples of photographs taken in various wavelengths. (**A**)—white light—full spectrum, (**B**)—blue light (450 nm), (**C**)—violet light (405 nm), (**D**)—blue and violet light together (405 + 450 nm). Green arrows indicate thin hyperkeratinized areas (brighter sites) which are not able to be seen in the classical white light examination (scale bar 10 mm).

**Figure 5 materials-14-05448-f005:**
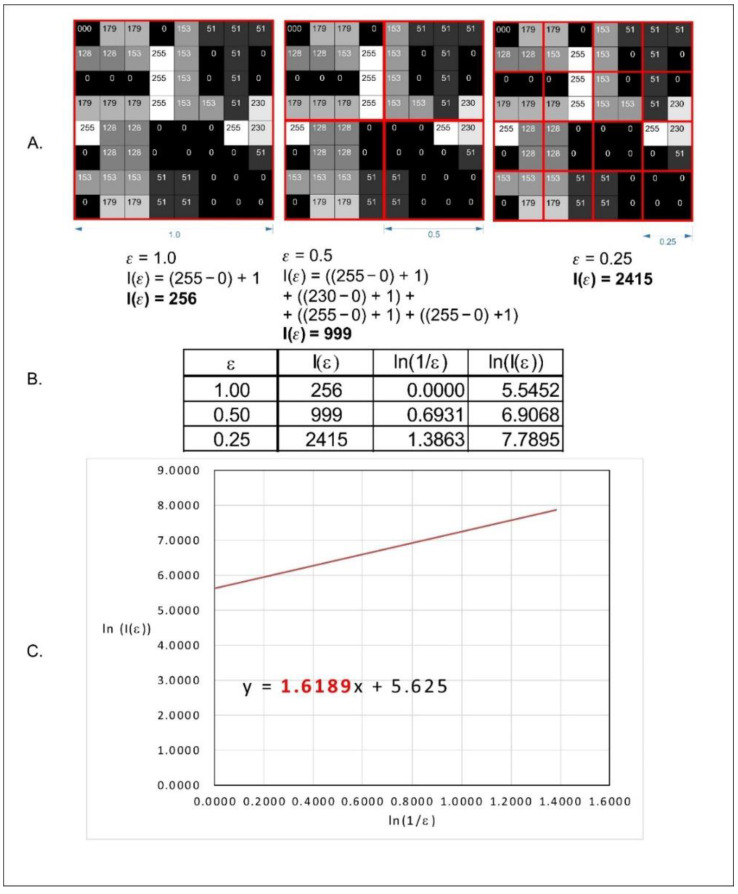
Graphical interpretation of the counting box method of fractal dimension counting. (**A**)—analysed bitmap, dimension of analysed square size (ε) (**B**)—Number of squares needed to cover the examined shape in the function of square size (ε). (**C**)—a straight line drawn through points from table B on the x-y chart in decimal logarithm scale. The slope factor of this straight line is the value fractal dimension counted by the box method [32].

**Figure 6 materials-14-05448-f006:**
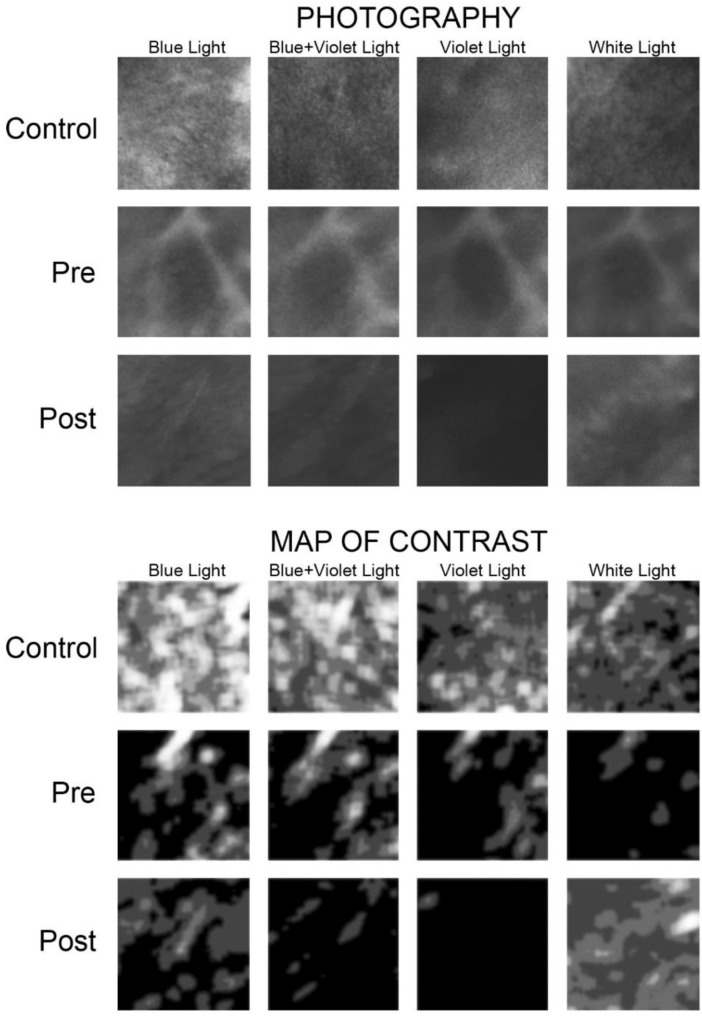
The upper panel consists of photographic images taken intra-orally under blue (450 nm), blue+violet (405 + 460 nm), violet (405 nm), and white light for control mucosa, lichen lesion pre-treatment, and lichen lesion post-treatment. The lower panel shows contrast distribution maps over the surface of the same photographs from the upper panel. White areas indicate areas of locally increased contrast in the image texture. On the contrary, dark areas indicate areas of low contrast (control—healthy mucous, Pre—lesion before treatment, Post—lesion after treatment).

**Figure 7 materials-14-05448-f007:**
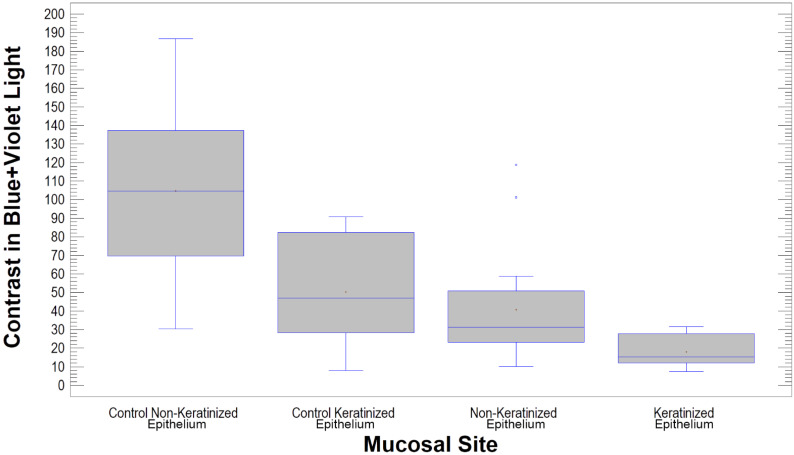
Pathological lesions in keratinized mucosa have significantly worse textural appearance than lesions formed in mucosa covered by non-keratinized epithelium (lower microcontrast value; *p* < 0.05). This is visible in the blue+violet light.

**Figure 8 materials-14-05448-f008:**
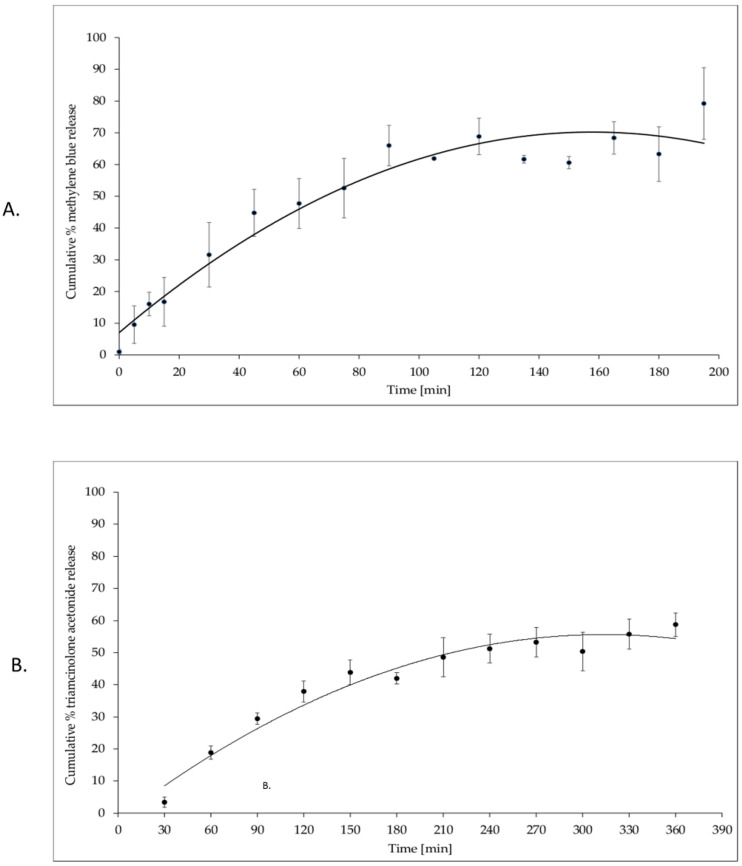
(**A**)—In vitro cumulative methylene blue release profile of from porous matrix containing 93.5 mg/cm^2^ methylene blue in dry mass. Results represent mean ± SE, n = 3. (**B**)—In vitro cumulative triamcinolone acetonide release profile of from porous matrix Pul40/Alg containing 0.05% substance in dry mass. Results represent mean ± SE, n = 3.

**Table 1 materials-14-05448-t001:** Methylene blue-loaded porous matrices composition.

Pullulan	Sodium Alginate	Glycerol	Methylcellulose	Methylene Blue
[% *w*/*w* in dry mass]				
27.0	7.3	38.9	4.3	22.5
[mg/cm^2^ of porous matrix]				
112	30.3	162	17.9	93.5

**Table 2 materials-14-05448-t002:** Porous matrices composition for triamcinolone application.

Batch Number	Pullulan	Sodium Alginate	Glycerol	Methylcellulose
	[% *w*/*w* in Dry Mass]			
PUL50/ALG	48.4	4.3	43.0	4.3
PUL40/ALG	36.6	9.8	48.8	4.9
PUL20/ALG	21.1	17.0	56.3	5.6

**Table 3 materials-14-05448-t003:** Results of paired Student *t*-test. Comparison size of surface before (Surf0) and after (Surf1) treatment for local steroid (steroid) and photodynamic therapy (PDT) treatment (SD—standard deviation, *t*—value of Student *t*-test, underlined—*p* < 0.05).

	Mean [mm^2^]	SD [mm^2^]	Difference [mm^2^]	SD Difference [mm^2^]	*t*	*p*
Steroid Surf0 (n = 27)	11.39	10.36	4.29	8.67	2.80	0.0087
Steroid Surf1 (n = 27)	7.10	8.09				
PDT Surf0 (n = 24)	18.39	16.74	7.34	8.70	4.05	0.0005
PDT Surf1 (n = 24)	11.04	11.80				

**Table 4 materials-14-05448-t004:** Results of unpaired *t*-Student test to comparison size of surface after treatment (Surf1) between local steroid (Steroid) and photodynamic therapy (PDT) treatment (SD—standard deviation, *t*—value of *t*-Student test).

vs.	Mean [mm^2^]	SD [mm^2^]	*t*	*p*
Steroid Surf1 (n = 27)	7.10	8.09	−1.47	0.1469
PDT Surf1 (n = 24)	11.04	11.80		

**Table 5 materials-14-05448-t005:** Results of paired *t*-Student test. Comparison between fractal dimension of lesion before treatment and healthy mucous in various wavelength (white—full spectrum of light, 405, 450, 405 + 450 nm), comparison between fractal dimension of lesion before and after treatment in various wavelengths (SD—standard deviation, *t*—value of Student *t*-test, underlined—*p* < 0.05).

	Fractal Dimension of:	Mean	SD	Difference	SD Difference	*t*	*p*
vs.	lesion before treatment in white illumination	1.589	0.110	0.107	0.095	−5.27	0.0000
	healthy mucous membrane in white illumination	1.696	0.033				
vs.	lesion before treatment in 450 nm illumination	1.491	0.137	0.122	0.135	−4.24	0.0004
	healthy mucous membrane in 450 nm illumination	1.613	0.093				
vs.	lesion before treatment in 405 nm illumination	1.504	0.133	0.215	0.118	−8.57	0.0000
	healthy mucous membrane in 405 nm illumination	1.718	0.051				
vs.	lesion before treatment in 405 + 450 nm illumination	1.473	0.133	0.196	0.137	−6.72	0.0000
	healthy mucous membrane in 405 + 450 nm illumination	1.668	0.064				
vs.	lesion before treatment in white illumination	1.589	0.110	0.001	0.123	0.04	0.9690
	lesion after treatment in white illumination	1.588	0.071				
vs.	lesion before treatment in 450 nm illumination	1.491	0.137	0.076	0.173	−2.08	0.0502
	lesion after treatment in 450 nm illumination	1.567	0.098				
vs.	lesion before treatment in 405 nm illumination	1.504	0.133	0.004	0.163	−0.12	0.9058
	lesion after treatment in 405 nm illumination	1.508	0.130				
vs.	lesion before treatment in 405 + 450 nm illumination	1.473	0.133	0.093	0.141	−3.08	0.0056
	lesion after treatment in 405 + 450 nm illumination	1.565	0.123				

**Table 6 materials-14-05448-t006:** The values of the Pearson correlation coefficient (r) between the value of fractal dimension (FD) calculated in different wavelengths and the size of the lesion before the treatment, Thongprasomn’s scale, the surface area of the lesion after treatment, the difference between its surface area after and before the treatment, and the percentage of lesion reduction (Thong—Thongprasom scale), underlined—|r| > 0.4.

FD of Lesion before Treatment vs.	W	B	V	B+V
Surface before treatment	0.2559	−0.4543	−0.1876	0.0836
Thong before treatment	0.0117	−0.1955	0.1292	0.2001
FD of lesion after PDT vs.	W	B	V	B+V
Surface after treatment	−0.411	−0.598	−0.748	−0.210
Surface after—surface before	0.137	0.202	−0.239	0.426
% reduction in size	−0.449	0.280	−0.837	−0.026
Thong after treatment	−0.402	0.243	−0.585	−0.299
FD of lesion after Steroid vs.	W	B	V	B+V
Surface after	0.049	0.226	0.214	0.273
Surface after—surface before	−0.580	−0.206	0.290	0.080
% reduction in size	−0.125	0.146	0.358	0.165
Thong after treatment	0.224	0.260	−0.172	0.317

**Table 7 materials-14-05448-t007:** Microcontrast analysis in lichen planus images in oral mucosa. Four ways of lesion illumination in pre-treatment time.

ROI	Blue Light	Blue+Violet Light	Violet Light	White Light
Control	75 ± 36	91 ± 47	118 ± 63	75 ± 34
Lesion	39 ± 30	35 ± 26	55 ± 44	43 ± 27
Sig.	*p* < 0.0001	*p* < 0.000005	*p* < 0.001	*p* < 0.001

Abbreviations: ROI—Region Of Interest; Sig.—Significance of Control versus Lesion differ.

**Table 8 materials-14-05448-t008:** Comparison of lesion texture (microcontrast) outcomes treated with PDT and steroid therapy.

Treatment	Blue Light		Blue+Violet Light		Violet Light		White Light	
	Pre	Post	Pre	Post	Pre	Post	Pre	Post
PDT	35 ± 26 †	41 ± 29 †	35 ± 30 †	49 ± 29 †	55 ± 51 †	39 ± 27 †	38 ± 19 †	31 ± 11
Steroid	45 ± 33 †	42 ± 25 †	35 ± 24 *†	70 ± 48 *†	54 ± 40 †	42 ± 38 †	48 ± 33 †	40 ± 28
Sig.	ns	ns	ns	ns	ns	ns	ns	ns

Sig.—Significance of PDT treatment versus Steroid treatment differ. There are no significant differences found in Pre- vs Post-Treatment in group PDT. *—only found significant increase in microcontrast after steroid therapy (*p* < 0.05). †—significantly lower value than in Control.

**Table 9 materials-14-05448-t009:** Physicochemical characteristics of the evaluated matrices. Data are presented as the mean ± SE. n = 3.

Physical Characteristics	Methylene Blue-Loaded Matrix	Matrices for Triamcinolone Acetonide		
		Pul50/Alg	Pul40/Alg	Pul20/Alg
Elongation (%)	142.91 ± 6.53	250.70 ± 25.00	189.10 ± 28.00	69.0 ± 8.30
Rupture force (g)	1498.70 ± 325.94	376.20 ± 36.70	253.20 ± 34.60	176.70 ± 100.46
pH	6.61 ± 0.23	6.84	6.85	6.82
Disintegration time (min)	>300	>300	>300	>400
Declareted drug content (%)	-	-	99.23 ± 5.29	-

## Data Availability

Data available from the authors: kamil.jurczyszyn@umed.wroc.pl; marcin.kozakiewicz@umed.lodz.pl.

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
