# Peer review of "Fractal Dimension and Texture Analysis of Lesion Autofluorescence in the Evaluation of Oral Lichen Planus Treatment Effectiveness"

_materials, 2021, doi:10.3390/ma14185448_

Round 1

Reviewer 1 Report

The authors described a comparison of photodynamic therapy versus topical steroids mediated with novel carriers in the treatment of oral lichen planus and estimation of treatment effectiveness. This article shows interesting results, but some aspects need to be improved before I recommend its publication in Materials.

  • Please define the abbreviations at their first use in the body of the manuscript, and in the abstract (ex. PDT).
  • Can the authors indicate a reference for the methodology described in lines 158-160?
  • I do not see the hyperkeratinized areas in Figure 4 as you indicated with arrows. I suggest to replace with images with higher resolution.
  • Please write the value of n replicates in the legend of figures where you have more replicates.
  • Discuss more the impact of you results for the material scientists & medical community.

Author Response

Dear Reviewer,

Thank you for your suggestions which improve our study.

  • Please define the abbreviations at their first use in the body of the manuscript, and in the abstract (ex. PDT).

Abstract is limited to 200 words so we used abbreviations to reduce amount of words. Now we added resolve of all abbreviations but this cause exceeding of abstract volume.

All abbreviations were resolved in the manuscript body.

  • Can the authors indicate a reference for the methodology described in lines 158-160?

We added 29 reference position: Fagerland, M.; Lydersen, S.; Laake, P. The McNemar test for binary matched-pairs data: mid-p and asymptotic are better than exact conditional BMC Med Res Methodol. 2013, 13, 13:91.

  • I do not see the hyperkeratinized areas in Figure 4 as you indicated with arrows. I suggest to replace with images with higher resolution.

Figure 4 is an example of autofluorescence of hyperkeratinized lesions in a various wavelengths. We used an example of the lesion where hyperkeratinized layer is too thin to be easy able to see in a classical white light examination. Even application blue or violet light offers us possibility to see border of the lesion which is wider. We added some information in the figure caption to be more clarify.

We will send this image in higher resolution as a separately file.

  • Please write the value of n replicates in the legend of figures where you have more replicates.

Corrected

  • Discuss more the impact of you results for the material scientists & medical community.

Corrected. We added one paragraph in discussion about it.

Manuscript was checked by certified translator. Certificate of translation was attached.

Best regards,

Authors.

Reviewer 2 Report

The paper entitled “A comparison of PDT versus topical steroids mediated with novel carriers in the treatment of oral lichen planus and estimation of treatment effectiveness using fractal dimension and texture analysis of lesion autofluorescence in various wavelengths” is interesting and has translational results.

Some improvements are suggested:

-INTRODUCTION: Authors have correctly described OLP and several studies on treatments and PDT. However, the authors have used matrix/scaffolds in their study, and they lacked to report about the clinical interaction between scaffolds and oral stem cells in tissue engineering and organs repair (Please, see “Tatullo M, Spagnuolo G, Codispoti B, Zamparini F, Zhang A, Esposti MD, Aparicio C, Rengo C, Nuzzolese M, Manzoli L, Fava F, Prati C, Fabbri P, Gandolfi MG. PLA-Based Mineral-Doped Scaffolds Seeded with Human Periapical Cyst-Derived MSCs: A Promising Tool for Regenerative Healing in Dentistry. Materials (Basel). 2019 Feb 16;12(4):597.”);

- More interestingly, authors should report something about the role of environment in MSCs behaviour, also in the light of their study, taking into consideration the concept of “multipotency” of bio-based scaffolds; they must also introduce something more about comparative cell models derived from dental tissues.

- Figures need scale-bars

- Figure 3 needs explaining in the legenda

-  Some typos throughout the text should be corrected

Minor suggestions

- Last figures may be included in single composite figures

- I would suggest to reduce title to “A comparison of photodynamic therapy versus topical steroids mediated with novel carriers in the treatment of oral lichen planus”

Author Response

Dear Reviewer,

Thank you for your suggestions which improve our study.

-INTRODUCTION: Authors have correctly described OLP and several studies on treatments and PDT. However, the authors have used matrix/scaffolds in their study, and they lacked to report about the clinical interaction between scaffolds and oral stem cells in tissue engineering and organs repair (Please, see “Tatullo M, Spagnuolo G, Codispoti B, Zamparini F, Zhang A, Esposti MD, Aparicio C, Rengo C, Nuzzolese M, Manzoli L, Fava F, Prati C, Fabbri P, Gandolfi MG. PLA-Based Mineral-Doped Scaffolds Seeded with Human Periapical Cyst-Derived MSCs: A Promising Tool for Regenerative Healing in Dentistry. Materials (Basel). 2019 Feb 16;12(4):597.”);

- More interestingly, authors should report something about the role of environment in MSCs behaviour, also in the light of their study, taking into consideration the concept of “multipotency” of bio-based scaffolds; they must also introduce something more about comparative cell models derived from dental tissues.

The carriers (matrices) of drugs which we used in our study are not the scaffold form. It is only a form of dressing which main advantage is a gluing to the mucous membrane. This feature enables to control topical drug administration (steroid and photosensitizer). Treatment methods which we applied did not based on multipotency of cells (especially MSCs derived from dental tissues) so we did not mentioned about it in introduction.

- Figures need scale-bars

It was added on figure 1 and 4.

- Figure 3 needs explaining in the legenda

Figure 3 shows disintegration test of matrices. More clarify caption was added.

-  Some typos throughout the text should be corrected

Corrected. Manuscript was checked by certified translator. Translation certificate was attached.

Minor suggestions

- Last figures may be included in single composite figures

The last two figures were integrated into one figure 8A and 8B.

- I would suggest to reduce title to “A comparison of photodynamic therapy versus topical steroids mediated with novel carriers in the treatment of oral lichen planus”

The first aim of our study was comparison of two treatment methods (PDT and topical sterorids) but the second aim was an application of modern mathematical methods of image analysis. This part gives a novelty aspect of our study. Fractal and texture analysis enabled to distinguish soft differences of lesions after treatment using various methods. So in our opinion second part of the title is necessary to underline novelty of analytic methods which we used.

Best regards,

Authors.

Round 2

Reviewer 2 Report

The authors have probably not well understood some of my previous suggestions. By the way, after an overall evaluation on the quality of this paper, I would give further suggestions to the authors.

Authors have reported their methods, referring a potential impact on OLP red lesions. An overall discussions on opportunities and limitations related to the other OLP forms should be added.

The authors only reported clinical and digitally analyzed images: no histological samples was investigated. Did the authors consider this potential bias?

Conclusions seem not properly reported in this pointed structure. Please, make them more readable with a narrative structure.

In figure 5: the authors have written erroneously some terms like "Fractal diemnsion "

The authors must revise figure 7: "keratonized"(?)

Furthermore, this reviewer still considers the title too long and it reports information usually highlighted in the abstract.

Scaffolds and carriers are often the same concepts, as scaffolds, matrices, and so on, may have the role of in situ releasing of active drugs or biological compounds. In this landscape, why did the authors choose this specific scaffolds/matrices?

Finally: the authors declare no conflict of interests; on the other side, they report to have a specific patent ( 9:223,123, B2; date of patent: 29 December 2015.). Could the authors please explain the role of this patent in this paper?

Author Response

Dear Reviewer,

  1. Authors have reported their methods, referring a potential impact on OLP red lesions. An overall discussions on opportunities and limitations related to the other OLP forms should be added.

In our study, we used the Thongsaprom scale to assess changes during therapy and cure the patient. It assumes that atrophic and erosive changes, i.e. the red replacement, are active foci of the disease into mucous and require treatment, while the white ones are treated like a healed form of OLP. That is why such a nomenclature is used in the discussion.

  1. The authors only reported clinical and digitally analyzed images: no histological samples was investigated. Did the authors consider this potential bias?

The main aim of this study was an application of fractal dimension and texture analysis based on clinical view of lesions (all lesions were histologically verified before treatment it is undelined in the materials and methods). Lichenoid lesion was diagnosed histologically following histological features that included irregular acanthosis, degeneration of the basal layer of the epithelium and a band of lymphohistiocytic infiltrate in the upper chorion composed almost exclusively of mature lymphocytes  (it is in the materials and methods section).

  1. Conclusions seem not properly reported in this pointed structure. Please, make them more readable with a narrative structure.

Narrative structure is in the results description.  

  1. In figure 5: the authors have written erroneously some terms like "Fractal diemnsion"

It was corrected in table 5.

  1. The authors must revise figure 7: "keratonized"(?)

The word has been corrected.

  1. Furthermore, this reviewer still considers the title too long and it reports information usually highlighted in the abstract.

Title was reduced to following form: Fractal Dimension and Texture Analysis of Lesion Autofluorescence in the Evaluation of Oral Lichen Planus Treatment Effectiveness.

  1. Scaffolds and carriers are often the same concepts, as scaffolds, matrices, and so on, may have the role of in situ releasing of active drugs or biological compounds. In this landscape, why did the authors choose this specific scaffolds/matrices?

As we mentioned in the introduction topically application of some drugs on the mucous membrane is very difficult. This problem is mostly highlighted in case of application of photosensitizer (PS) during photodynamic therapy. Some of factors in oral cavity have an influence of topically drug administration, for example continuously movements of muscles and saliva flowing.  Matrices which offer precisely estimate of PS and steroid administration was used in our study to achieve similarity.

  1. Finally: the authors declare no conflict of interests; on the other side, they report to have a specific patent ( 9:223,123, B2; date of patent: 29 December 2015.). Could the authors please explain the role of this patent in this paper?

One author of this study is an author of mentioned patent number (this information was added to the manuscript). The laser device which was used in our study was built on the base of this patent. We removed number of patent in materials and section chapter. The patent describes the method of coupling multiple laser diodes into a single fiber. We have used this method for constructing our laser source but we have no commercial interest in this device. It is only used for research purposes. .  It is no conflict of interest as we mentioned. Best regards,Authors.